# Stabilization of F-Actin Cytoskeleton by Paclitaxel Improves the Blastocyst Developmental Competence through P38 MAPK Activity in Porcine Embryos

**DOI:** 10.3390/biomedicines10081867

**Published:** 2022-08-02

**Authors:** Seung-Yeon Joe, Seul-Gi Yang, Jae-Ho Lee, Hyo-Jin Park, Deog-Bon Koo

**Affiliations:** 1Department of Biotechnology, College of Engineering, Daegu University, 201 Daegudae-ro, Jillyang, Gyeongsan 38453, Korea; zoey3997@gmail.com (S.-Y.J.); foreverday37@naver.com (S.-G.Y.); 2Institute of Infertility, Daegu University, 201 Daegudae-ro, Jillyang, Gyeongsan 38453, Korea; 3Department of Biomedical Science, College of Life Science, CHA University, Pocheon 11160, Korea; jaeho@cha.ac.kr; 4CHA Fertility Center, Seoul Station, Hangang-daero, Jung-gu, Seoul 04637, Korea

**Keywords:** paclitaxel, F-actin, cytoskeleton, adducin, P38 MAPK, porcine embryo

## Abstract

Changes in F-actin distribution and cortical F-actin morphology are important for blastocyst developmental competence during embryogenesis. However, the effect of paclitaxel as a microtubule stabilizer on embryonic development in pigs remains unclear. We investigated the role of F-actin cytoskeleton stabilization via P38 MAPK activation using paclitaxel to improve the developmental potential of blastocysts in pigs. In this study, F-actin enrichment and adducin expression based on blastomere fragment rate and cytokinesis defects were investigated in cleaved embryos after in vitro fertilization (IVF). Adducin and adhesive junction F-actin fluorescence intensity were significantly reduced with increasing blastomere fragment rate in porcine embryos. In addition, porcine embryos were cultured with 10 and 100 nM paclitaxel for two days after IVF. Adhesive junction F-actin stabilization and p-P38 MAPK activity in embryos exposed to 10 nM paclitaxel increased significantly with blastocyst development competence. However, increased F-actin aggregation, cytokinesis defects, and over-expression of p-P38 MAPK protein by 100 nM paclitaxel exposure disrupted blastocyst development in porcine embryos. In addition, exposure to 100 nM paclitaxel increased the misaligned *α*-tubulin of spindle assembly and adhesive junction F-actin aggregation at the blastocyst stage, which might be caused by p-P38 protein over-expression-derived apoptosis in porcine embryos.

## 1. Introduction

Paclitaxel, also known as taxol, possesses a unique ability to promote microtubule cytoskeleton stabilization in mammalian cells [1,2] and typically promotes the interference of the regular dynamic reorganization of the microtubule network required for mitotic cellular functions [3]. In addition, paclitaxel is utilized as an antitumor drug via interference with the mitotic spindle in female ovarian and breast cancers [4]. As typical mitogen-activated protein kinase (MAPK) markers, both extracellular signal-regulated kinase (ERK) and P38 MAPK cascades are essential for the apoptotic response to paclitaxel-induced cell death in most cancer cells [5]. In particular, P38 MAPK contributes to microtubule disassembly via cytosolic microtubule-binding protein phosphorylation [6,7]. In mice, paclitaxel is involved in the stabilization of microtubules related to spindle assembly, which can protect zygotes at risk for aneuploidy from freeze-thawed embryos [8]. Paclitaxel may make up for microtubules in spindle assembly during cell division, but its regulatory mechanisms for promoting blastocyst developmental potential have not yet been elucidated in porcine preimplantation embryos in vitro.

Importantly, fibrous actin (F-actin), a major cytoskeleton component, is stabilized by paclitaxel [9]. Cytoskeletal F-actin assembly and microtubule reorganization are the principal cellular events responsible for changes in the shape of the cleaved embryo that determine the blastocyst developmental potential. The cytoskeleton provides morphological stability in the oocyte and plays a crucial role in dynamic cell shape changes and the facilitation of cell motility [10,11]. During the cleavage stages in pre-implantation embryos, the actin cytoskeleton is deeply connected to driving cell shape changes, tissue bands, and fusion events that characterize embryogenesis [12]. Interactions between dynamic microtubules and actin filaments, known as F-actin, underlie a range of cellular processes, including cell polarity and motility [13]. Adducin is the only actin-binding protein and comprises three closely related components (α, β, and γ), where α and γ are ubiquitously expressed in mammals [14]. Adducin has a major function in regulating the actin cytoskeleton to cap the barbed ends of F-actin. In particular, adducin regulates the actin cytoskeleton by promoting the bundling of actin filaments and inhibiting the incorporation of new actin monomers [15]. Despite the beneficial effects of F-actin morphology-mediated microtubule stabilization, the interaction of α-adducin with preimplantation embryo development has not been defined in pigs.

The oocyte-to-embryo transition after fertilization requires various cytoskeletal changes, such as reorganization of the oocyte cortex and preparation for blastomere division [16]. In addition, early cleavage events in embryos play a vital role in determining embryonic developmental fate, including blastocyst formation [17]. Many studies have shown that poor-quality embryos have problems, such as blastomere fragment percentage [18], cytokinesis defects [17], and microtubule formation [19] at the cleavage stage after IVF. A previous study has shown that F-actin disorganization is implicated in impaired development of in vitro produced (IVP) embryos during the preimplantation stage [20]. Additionally, Natale et al. [21] described that P38 MAPK activity is required to support developmental capacity in pre-implantation mouse embryos. In addition, activated P38 MAPK is involved in the microtubule-dependent spindle assembly and stabilization of F-actin, which plays an essential role in cytokinesis.

Although paclitaxel features as an F-actin cytoskeleton and microtubule stabilizer with high potency against cancer cells, the effects of paclitaxel on the cleaved embryo process and blastocyst developmental capacity in porcine embryos still need to be explored. First, we investigated the relationship between embryo quality and F-actin/adducin expression patterns in porcine embryos according to blastomere fragmentation and cytokinesis defect percentage divided into three groups (Groups A, B, and C) during in vitro culture (IVC). In this study, we assessed the role of paclitaxel that the F-actin cytoskeleton may play in determining morphological properties of equal division, enrichments, and aggregation in blastomere of the porcine embryos during the cleaved stages. Second, porcine embryos were cultured in vitro and treated with different paclitaxel concentrations for two days after IVF. We further analyzed the effects of paclitaxel treatment on F-actin morphology, aggregation, adducin protein levels, and P38 MAPK activation in porcine embryos to identify microtubule and F-actin cytoskeleton stabilization-associated mechanisms for embryonic developmental competence.

## 2. Materials and Methods

### 2.1. Chemicals

All chemicals used in this study were purchased from Sigma Chemical Co. (St. Louis, MO, USA) unless otherwise specified.

### 2.2. Oocyte Collection and In Vitro Maturation (IVM) of Porcine Oocytes

Porcine ovaries were obtained from a local slaughterhouse and transported to the laboratory in 0.9% saline (*w*/*v*) supplemented with 75 μg/mL penicillin G at 38.5 °C within 2–3 h. Immature cumulus–oocyte complexes (COCs) were aspirated from medium-sized (3–6 mm) follicles with an 18-gauge needle attached to a 10 mL syringe. Immature COCs with several layers of cumulus cells and a dark cytoplasm were selected for maturation. Approximately 50–60 immature COCs were allowed to mature in an IVM medium in four-well multi-dishes (500 μL/well) (Nunc, Roskilde, Denmark) maintained in culture under 5% CO_2_ air atmosphere at 38.5 °C for 22 h. A North Carolina State University-23 (NCSU-23; pH 7.4) medium supplemented with 10 IU/mL pregnant mare’s serum gonadotropin (PMSG), 10 IU/mL human chorionic gonadotropin (hCG), 0.57 mM cysteine, 10 ng/mL *β*-mercaptoethanol, 10 ng/mL epidermal growth factor (EGF), and 10% follicular fluid (*v*/*v*) were used for oocyte maturation. After the oocytes were cultured for 22 h, the COCs were further cultured in a hormone-free IVM medium for an additional 22 h under a 5% CO_2_ air atmosphere at 38.5 °C.

### 2.3. In Vitro Fertilization (IVF) and Culture (IVC)

IVF was conducted as described in a previous study [22]. Briefly, the fertilization medium used a modified Tris-buffered medium (mTBM; pH 9.5) consisting of 7.5 mM CaCl_2_, 113.1 mM NaCl, 3 mM KCl, 11 mM glucose, 20 mM Tris, 5 mM sodium pyruvate, 2.5 mM caffeine sodium benzoate, and 1 mg/mL bovine serum albumin (BSA). Fresh semen was supplied by the Darby Porcine AI Center (Anseong, Korea) twice a week and stored at 17 °C. The semen was washed three times by centrifugation with phosphate-buffered saline (PBS; pH 7.4) supplemented with 0.37 mg/mL penicillin G, 0.03 mg/mL streptomycin sulfate, 0.1 mg/mL CaCl_2_, and 1 mg/mL BSA. After the final wash, spermatozoa were resuspended in mTBM at pH 7.8. At the end of the maturation period, the cumulus cells adhering to the oocyte were removed using a TL-HEPES (pH 7.4) medium supplemented with 0.1% hyaluronidase. Denuded oocytes were transferred into a 48 μL drop of mTBM under mineral oil. Then, diluted spermatozoa (2 μL) were added to a 48 μL drop containing 15–20 oocytes to give a final concentration of 1.5 × 10 ^5^ sperms/mL for 6 h under 5% CO_2_ air atmosphere at 38.5 °C. After 6 h of co-incubation, the embryos were washed three times with an IVC medium (PZM-3 containing 3 mg/mL bovine serum albumin; pH 7.4). Embryos were cultured in a 50 μL drop of non-treated IVC medium (control) and IVC medium supplemented with paclitaxel (10 and 100 nM). After 48 h, cleaved embryos were further cultured in a non-treated 50 μL drop of IVC medium for four days. Blastocyst formation was acquired and assessed after 6 days of culture. The evaluation of blastocysts was classified into four types (Early, Mid, Late, and Expanded) according to morphology at the acquisition time. The early blastocyst has a small size, indistinct inner cell mass (ICM), and generation of trophoblast. The mid-blastocyst stage has a middle size, incomplete ICM, and distinct trophoblast. The late blastocyst has a large size, distinct ICM, and trophoblast. The expanded blastocyst has a huge size, distinct ICM, and trophoblast.

### 2.4. Evaluation of Cleaved Embryos According to the Blastomere Fragmentation

Blastomere fragmentation in cleaved embryos was evaluated under an optical microscope, as previously described [23,24]. We defined three groups (Group A, B, and C) according to the morphology of cleaved embryos on the 2 days after fertilization during early embryonic development. Group A was selected from cleaved embryos that have complete division and no blastomere fragmentation. Group B was classified as minor fragmentation and has <15% blastomere fragmentation. Cleaved embryos with >25% fragments were distinguished as Group C and showed failures of mitotic division. The percentage of blastomere fragmentation was visually determined by the extent of fragments occupying the total area.

### 2.5. Filamentous Actin and Immunofluorescence (IF) Staining

FITC-Phalloidin (P5282, Sigma, St. Louis, MO, USA) was prepared according to the manufacturer’s instructions. Briefly, embryos were fixed with 3.7% formaldehyde overnight at 4 °C and permeabilized with 0.5% Triton X-100 (*v*/*v*) for 30 min at room temperature. The washed embryos were blocked with 1% BSA and 0.1% polyvinyl alcohol (PVA) in PBS for 1 h at room temperature. Blocked embryos were incubated with phalloidin (5 μg/mL), mouse monoclonal antibody adducin *α* (Santa Cruz, 1:100, sc-133079, Santa Cruz, Dallas, TX, USA), and mouse monoclonal FITC-conjugated anti-α-tubulin antibody (1:100) overnight at 4 °C. After primary antibody incubation, the embryos were incubated with the secondary antibody, Alexa Fluor 555 goat anti-rabbit IgG (1:200 dilution) (Thermo Scientific, Waltham, MA, USA). Incubated embryos were counterstained with 1.5 μg/mL 4′,6-diamidino-2-phenylindole (DAPI; Vector Laboratories, Burlingame, CA, USA) for 10 min and mounted on glass slides. Images were acquired using an LSM 800 confocal microscope (Zeiss, Jena, Germany). F-actin and adducin intensities and length were measured by green and red fluorescence quantification in embryos using ImageJ 1.46r software (NIH, Bethesda, MD, USA). Excessive F-actin aggregation was defined as F-actin thickness larger than 6 μm, and the number of embryos including the F-actin aggregation among the total embryos was indicated as a percentage.

### 2.6. Assessment of Cellular Apoptosis in Porcine Blastocysts

Apoptosis was detected by terminal deoxynucleotidyl transferase-mediated dUTP nick-end labeling (TUNEL) using an in situ cell death detection kit (Roche Diagnostics, Mannheim, Germany), according to the manufacturer’s protocol. Blastocysts were permeabilized with 0.5% Triton X-100 (*v*/*v*) for 30 min at room temperature. The fixed blastocysts were incubated in the TUNEL reaction medium at 38.5 °C for 1 h. After washing three times with 0.1% PVA in PBS for 10 min, the embryos were counterstained with 1.5 μg/mL DAPI (Vector Laboratories) for 10 min and mounted on glass slides. Images were acquired using the iRiS^TM^ digital cell imaging system (Logos Biosystems). We measured the number and rate of TUNEL-positive cells (apoptotic cells) in the total nuclei per blastocyst through the acquired images.

### 2.7. Protein Extraction and Western Blot Analysis

Cleaved embryos and blastocysts (70 embryos per sample) were lysed using PRO-PREP protein lysis buffer (iNtRON, Daejeon, Korea), and the lysates were separated on an 8–12% sodium dodecyl sulfate–polyacrylamide gel electrophoresis (SDS-PAGE) gel. The isolated proteins were transferred onto a pure nitrocellulose membrane (Pall Life Sciences, Port Washington, NY, USA). After overnight blocking with 5% non-fat skim milk at 4 °C, the membranes were incubated with primary antibodies against anti-P38 MAPK (Santa Cruz), adducin (Santa Cruz), and β-tubulin (Santa Cruz). In addition, the membranes were incubated with secondary HRP-conjugated anti rabbit/mouse IgG (Thermo Fisher Scientific, MA, USA) for 1 h at room temperature and washed with Tris-buffered saline containing 0.1% Tween 20 (TBST) buffer. Antibody binding was detected using an ECL kit (Bio-Rad, Hercules, CA, USA). The bands were visualized using the Fusion Solo software (Vilber Lourmat, Collégien, France). The band intensities were quantified by densitometry using ImageJ software (NIH, Bethesda, MD, USA). Original blot images are included in the Appendix A.

### 2.8. Statistical Analysis

All the experiments in triplicate are presented as the mean ± standard deviation (SD). All results were analyzed using t-tests or one-way analysis of variance (ANOVA) followed by Tukey’s multiple comparison test. All data were analyzed using the GraphPad Prism software (version 5.0; San Diego, CA, USA). Histogram values of densitometry were evaluated using the ImageJ software (NIH, Bethesda, MD, USA). Differences are considered significant at * *p* < 0.05, ** <0.01, and *** <0.001.

## 3. Results

### 3.1. Investigation of Actin Filament Organization Changes in Porcine Embryos by Morphological Classification at the Cleavage Stage

Previously, the percentage of blastomere fragments was suggested as a criterion for embryo quality evaluation after IVF [25,26]. In this study, we divided the cleaved embryos into three groups (Group A: no fragment, Group B: <15% fragments, and Group C: >25% fragments) according to blastomere fragmentation 48 h after IVC (Figure 1A,B). The percentage of cleaved embryos was significantly higher (*p* < 0.05) in group A (58.9 ± 5.0%) than in groups B (23.7 ± 7.9%) and C (17.4 ± 7.4%) (Figure 1C and Table 1). We investigated the F-actin fluorescence expression pattern using FITC-phalloidin staining in the cleaved embryos of Groups A, B, and C. According to the increasing blastomere fragment rate in cleaved embryos, the F-actin filament organization and fluorescence intensity at the adhesive junction site significantly reduced in other groups compared to Group A (Group B: *p* < 0.05, and Group C: *p* < 0.001, Figure 1D,E). We also investigated the co-expression of F-actin and adducin in cleaved embryos from Groups A, B, and C (Figure 1F–H). Because adducin functions as an actin-binding protein for fixing an F-actin filament, we first investigated the co-localization of F-actin (green) and adducin (red) fluorescent expression in porcine embryos from Groups A, B, and C. Adducin fluorescence intensity significantly decreased (Figure 1H) with increasing cytokinesis defects in embryos, and the F-actin aggregation ratio significantly increased in cleaved embryos of Group C compared with other groups (Figure 1G). However, co-expression of adducin and F-actin was not evidently observed in any of the groups (data not shown). These results indicated that adhesive junction F-actin enrichment and actin-binding protein (adducin) expression were reduced according to abnormal morphological blastomere content in cleaved embryos after IVF.

### 3.2. Interrelation of Blastomere Fragmentation and Cleaved Embryo Ratio in Paclitaxel Exposed Porcine Embryos

Some studies have reported that actin cytoskeleton organization and blastomere morphology are crucial factors that contribute to embryo quality [26]. Thus, we investigated the percentage of blastomere fragments based on equal blastomere division and fragment rate in porcine cleaved embryos two days after treatment with 10 or 100 nM paclitaxel. Embryos exposed to 10 nM paclitaxel showed the highest proportion of Group A embryos with 0% blastomere fragmentation (Figure 2B and Table 2). These findings showed similar patterns for cleaved embryo ratio in 10 nM paclitaxel treated embryos belonging to Groups A, B, and C compared with results of non-treated porcine embryos (*p* < 0.05; Group A: 57.6 ± 9.9% vs. Group B: 23.6 ± 6.4%, and C: 18.8 ± 8.8%). We next explored whether the application of 100 nM paclitaxel affected the embryonic cleavage percentage in pigs during IVC (Figure 2H and Table 2). Paclitaxel treatment with 100 nM showed that the overall proportion of Groups B and C with increased blastomere degradation or fragmentation was higher than that of Group A (1.57-fold, *p* < 0.05; Group A: 29.8 ± 8.3% vs. Group C: 46.7 ± 8.5%).

### 3.3. Co-Expression of F-Actin/Adducin and Adhesive Junction F-Actin Enrichment According to Blastomere Fragmentation by Paclitaxel Exposure in Porcine Embryos

Paclitaxel treatment (10 or 100 nM) for two days after IVF showed a significant difference in decreasing adhesive binding F-actin intensity from Groups B and C compared to Group A (Paclitaxel 10 nM: *p* < 0.05 and 100 nM: *p* < 0.01; Figure 2C,I). We explored whether paclitaxel exposure (10 or 100 nM) affected the actin cytoskeleton of cleaved embryos, including cytokinesis defects, by measuring F-actin and adducin fluorescent expression (Figure 2D–F,J–L). As shown in Figure 2, F-actin aggregation in cleaved embryos after paclitaxel treatment was significantly increased (paclitaxel 10 nM: *p* < 0.05 and 100 nM: *p* < 0.01) in Group C compared to that in Group A. Adducin fluorescent expression was significantly reduced in Group C compared to that in Group A after paclitaxel exposure with 10 (*p* < 0.01) or 100 nM (*p* < 0.001). F-actin aggregation was significantly increased in cleaved porcine embryos in Groups B and C with or without paclitaxel treatment. Figure 2D,J showed accompanied decreasing fluorescence expression of adducin in Group C, including blastomere and cytokinesis defects, irrespective of paclitaxel treatment concentration.

### 3.4. Blastocyst Developmental Competence and Quality by Paclitaxel Treatment in Porcine Embryos

Actin filaments are involved in the cleavage process via F-actin expansion before blastocyst formation, and they can affect the spindle assembly checkpoint and spindle positioning in oocytes or embryos. To investigate whether microtubule stabilization by paclitaxel is related to blastocyst developmental competence and cellular apoptosis, porcine embryos were cultured for two days in a 10 or 100 nM paclitaxel-supplemented medium and cultured for four days without paclitaxel (Figure 3 and Table 3). After exposure to 10 nM paclitaxel, the blastocyst development rate (control: 22.2 ± 6.0% vs. 10 nM paclitaxel: 29.4 ± 4.7%; *p* < 0.05) and expanded blastocyst formation were significantly increased (*p* < 0.05, Figure 3A–D) when compared with the other groups. However, blastocyst developmental competence and expanded blastocyst formation were significantly reduced in developed blastocysts from 100 nM paclitaxel-treated embryos (*p* < 0.01; 100 nM paclitaxel: 13.5 ± 4.3%). Furthermore, DNA apoptosis in blastocysts exposed to 100 nM significantly increased (*p* < 0.01; Figure 3E–G and Table 4). To indicate that the paclitaxel-induced improvement in in vitro embryonic development partly depends on microtubule-modulating roles, we investigated the changes in α-tubulin-mediated spindle formation in the paclitaxel-treated group. Interestingly, as a result of our measurements of average microtubule bundle length (μm) and abnormal spindle rate in the paclitaxel-treated group, only 100 nM paclitaxel-exposed embryos were significantly increased (*p* < 0.05, Figure 3H–K). The misaligned chromosome rate decreased (*p* < 0.05) in blastocysts exposed to 10 nM paclitaxel compared with that in the other groups (Figure 3J). These results suggest that exposure to 10 nM paclitaxel can improve blastocyst developmental competence and quality through microtubule stabilization related to spindle assembly.

### 3.5. Correlation with F-Actin Organization Changes by Paclitaxel in Porcine Embryos during IVC Progression

The possibility that paclitaxel is involved in microtubule stabilization led us to hypothesize that IVP embryos are involved in blastomere division via actin organization. Thus, we performed a FITC-phalloidin staining analysis to study the distribution of F-actin in cleaved (Figure 4) and blastocyst (Figure 5) porcine embryos. Impaired blastocyst developmental competence by 100 nM paclitaxel treatment can disrupt the F-actin structure in porcine embryos at the cleavage and blastocyst stages. An increase in excessive F-actin aggregation was observed (*p* < 0.05) in embryos treated with 100 nM paclitaxel at the cleavage (Figure 4A,B) and blastocyst stages (Figure 5A,B). In addition, F-actin fluorescent expression showed solidity at the adhesive bonding sites of cleaved embryos and blastocysts in the 10 nM paclitaxel-treated group (Figure 4C and Figure 5C). These results suggested that paclitaxel 10 nM treatment of porcine embryos for two days after IVC improves the stabilization of tubulin and assists in enhancing F-actin in cleaved embryos and blastocysts.

### 3.6. F-Actin Stabilization Induced by Paclitaxel Exposure Activates P38 MAPK in Porcine Embryos at the Cleaved and Blastocyst Stages

Paclitaxel exposure (10 or 100 nM) can affect F-actin organization stability through P38 MAPK activation [26], and cleaved embryos can require P38 MAPK activity to improve blastocyst developmental capacity [21]. Hence, we next explored whether paclitaxel exposure affects P38 activation (p-P38 and P38) and adducin protein levels in porcine embryos at the cleavage and blastocyst stages, respectively. As shown in Figure 4, the p-P38 and adducin protein levels in embryos treated with only 10 nM at the cleavage stage were higher (*p* < 0.05) than those in the other groups. Total P38 protein expression in porcine embryos was not significantly different between the paclitaxel-treated groups and the control at the cleavage stage. Finally, we investigated P38 MAPK activation related to F-actin stabilization in blastocysts after paclitaxel exposure (Figure 5D–G). Although the p-P38 MAPK protein level in 10 nM paclitaxel-exposed blastocysts significantly diminished (*p* < 0.05), total P38 MAPK protein expression did not change with or without paclitaxel treatment. Thus, our data showed that only the 10 nM paclitaxel-treated group was significantly altered (Figure 5G). These data suggest a close association between F-actin structure stability and P38 MAPK activity and embryonic developmental capacity in porcine embryos.

## 4. Discussion

The effects of the F-actin cytoskeleton on embryonic developmental potential have been relatively well studied; however, the mechanism by which stabilization of actin filaments and microtubules by paclitaxel is involved in blastomere fragmentation or cytokinesis defects in porcine embryos is largely unknown. In the current study, we demonstrated the correlation of the developmental capacity to the blastocyst stage in porcine embryos through F-actin morphology and enrichment in the adhesive region of blastomeres according to the abnormal division percentage of blastomeres. Our findings indicate that treatment with 10 nM paclitaxel plays an important role in blastocyst development and quality via modulation of F-actin organization and microtubule stabilization by P38 MAPK activity in porcine embryos. Therefore, our results suggested that the paclitaxel can reliably improve porcine early embryonic developmental capacity via actin cytoskeleton stabilization by F-actin enrichment, adducin protein levels, and P38 MAPK activation during preimplantation development in vitro.

The standard morphological parameters for embryo quality assessment include blastomere size, degree of fragmentation, and cytoplasm appearance [27]. In addition, the actin filament as part of the cytoskeleton plays a crucial role in cell division and cell shape maintenance [28]. A previous study showed that actin filament distribution is closely connected in developing pig embryos [29]. Moreover, the F-actin distribution during the whole process of the Drosophila cleavage embryo was associated with the newly formed plasmalemmas along their length [30]. Additionally, excessive F-actin aggregation impedes mitosis, leading to cytokinesis failure in human fibrosarcoma cells [31]. Based on a previous study, we confirmed that actin filament (F-actin) distribution and microtubule (similar to *α*-tubulin) morphology could affect blastocyst developmental competence in vitro [32]. Our results indicated that abnormal morphological embryos and blastomere fragment percentages were applied as appraisal standards for further embryonic developmental competence during the cleavage stages (Figure 6A).

Coordination of F-actin and microtubule dynamics is important for cellular motility and division. Adducin (known as ADD1) is an essential molecule for maintaining a wide range of physiological functions, including cytoskeleton and actin dynamics [33]. Adducin is a family of cytoskeleton proteins encoded by three genes (α, β, and γ). Especially, *α*-adducin is located in the spectrin–actin junction of the membrane cytoskeleton and plays a role as a cytoskeletal protein that modulates Na-K pump activity and sodium homeostasis [34,35]. A recent study showed that downregulation of adducin attenuates the formation of f-actin bundles [36]. In addition, adducin knock-out neurons induce an unstable actin cytoskeleton and microtubule dynamics [37]. Although the F-actin mechanism controls spindle assembly in mouse zygotes, little is known about the mechanisms underlying porcine embryonic developmental competence. In this study, we showed a decrease in F-actin fluorescence intensity (Figure 1D) and adducin fluorescence expression with increased F-actin aggregation (Figure 1F) in cleaved embryos as the content of blastomere fragmentation increased. In addition, as the blastomere fragmentation ratio increased, the results of the co-expression intensities of adducin and F-actin showed no significant difference between each group (data not shown). We suppose that this result is because the fluorescence expression intensity of F-actin and adducin decreases simultaneously in Groups B and C compared to Group A. These findings demonstrate a direct correlation between morphology of abnormal cleaved porcine embryos and F-actin aggregation according to the percentage of blastomere fragments.

As an anticancer drug, paclitaxel is one of the most common chemotherapeutic agents used to treat breast, non-small cell lung, and ovarian cancers [38]. Paclitaxel is a well-known microtubule stabilizer that selectively arrests the cell cycle at the G2/M phase [39]. A previous study showed the interplay between the induction of microtubule stabilization and F-actin dynamics by paclitaxel treatment in neuronal cells [40]. Additionally, paclitaxel positively affects the development of morula/blastocyst stages and improves the developmental potential of vitrified human and mouse oocytes [8]. Our findings demonstrated that treatment with 10 nM paclitaxel improved the developmental efficiency of blastocysts, while the proportion of embryos associated with blastomere fragments was not significantly different from that in the non-treated group (Figure 2 and Figure 3).

In the cleavage stage of embryos, F-actin can affect the equal division of blastomeres and spindle assembly-related microtubule conditions [41]. Interestingly, the functional effects of 10 nM paclitaxel as a microtubule stabilization agent improved the expression of F-actin at the joint sites of cleaving blastomeres. Enrichment of F-actin enhances the stability of spindle formation during cytokinesis and plays a vital role in chromosome segregation during the cell cycle [42]. The results shown in Figure 3K confirmed that 10 nM paclitaxel treatment maintained spindle microtubule bundle length at the blastocyst stage. However, paclitaxel was found to induce cytotoxicity during treatment periods in a concentration-dependent manner [40]. As expected, treatment with 100 nM paclitaxel interrupted blastocyst development and quality (Figure 3). Prior to apoptosis, paclitaxel treatment strongly activated ERK and P38 MAPK in MCF7 cells, and P38 MAPK is essential for paclitaxel-induced cell death [43]. The activity of P38 MAPK signaling by paclitaxel stimulation was responsible for the sensitivity to F-actin reorganization for the stability of the cytoskeleton structure [9]. Although P38 MAPK is associated with apoptosis, P38 MAPK signaling is indispensable during mouse preimplantation development. During mouse early embryo development, P38 MAPK expression increases but decreases as it progresses to the blastocyst stage [44]. As shown in Figure 4, we confirmed the response to 100 nM paclitaxel-derived cellular toxicity focused on F-actin filament disruption through F-actin aggregation (Figure 4A–C) in porcine embryos at the cleavage stage. The high expression of phosphorylated P38 MAPK protein (p-P38) probably improved blastocyst developmental competence in porcine embryos. P38 MAPK signaling is involved in F-actin cytoskeleton and reorganization [45] and activated P38 protein promotes embryonic developmental capacity [46]. In our study, the expression level of p-P38 MAPK was significantly reduced in blastocysts derived from embryos treated with 10 nM paclitaxel treated embryos (Figure 5D). These findings indicate that P38 MAPK activity by paclitaxel exposure is required for the division process of equal blastomeres accompanied by F-actin enrichment and structural stabilization during the cleavage stage of porcine embryos (Figure 6B). We also performed Western blot analysis of blastocysts developed from embryos already exposed to paclitaxel. As shown in Figure 5, these results imply that the increasing P38 MAPK activation by paclitaxel at the cleavage stage may induce further blastocyst formation and development rate with assistance from F-actin morphology stability in pigs.

## 5. Conclusions

In summary, our findings indicate that actin cytoskeleton morphology and organization are essential to preserve the equal division of blastomere through F-actin and actin binding protein (adducin) in porcine embryos during the cleavage stage of IVC. Therefore, our study provides new evidence for the blastocyst developmental potential by F-actin expression intensity and F-actin aggregation in porcine cleaved embryos. Especially, our results showed the positive effect of 10 nM paclitaxel on actin cytoskeleton morphology and organization associated with the P38 MAPK activity in porcine embryo development. We demonstrated that the toxicity effect of 100 nM paclitaxel exposure leads to reduction of blastocyst developmental competence by disruption of F-actin cytoskeleton. Therefore, control of F-actin aggregation, intensity, and microtubule stabilization by 10 nM paclitaxel exposure might apply for an evaluation method of the blastocyst developmental competence potential through embryo quality and equal dividing blastomere in porcine embryos in vitro. Moreover, these results contributed towards understanding the roles of paclitaxel on embryonic developmental potential by F-actin skeleton stabilization at the adhesive junction site of cleaved embryos in pigs.

## Figures and Tables

**Figure 1 biomedicines-10-01867-f001:**
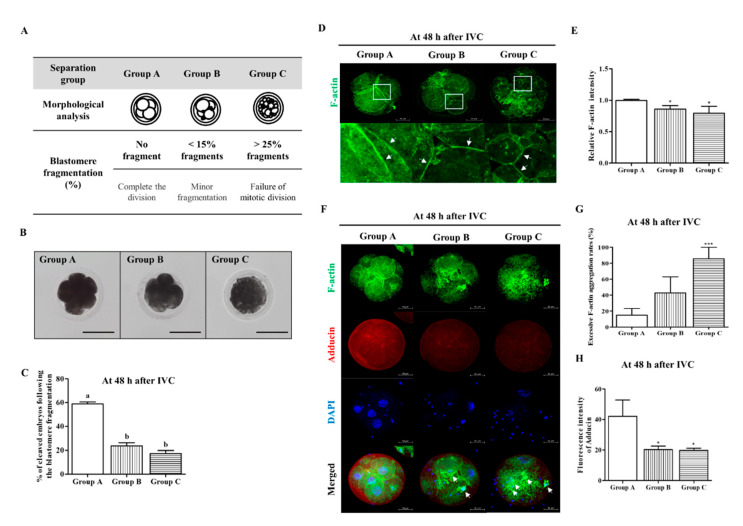
Porcine embryos reduced accumulation of adhesive junction F-actin and adducin expression according to blastomere fragment rate. (**A**) Typical images were divided into three groups (Groups A, B, and C) based on morphological criteria of blastomere in porcine embryos at the cleavage stage. Group A: Equal size blastomeres and completed the division state without fragmentation. Group B: Minor fragmentation state with less than 15%. Group C: Unequal size blastomeres with more than 25% fragments (failure of mitotic division state). Scale bar = 150 μm. (**B**) Bright-field micrograph of porcine embryos (4- or 8-cell stage) in three groups (Groups A, B, and C) according to the presence or absence of blastomere fragmentation. (**C**) Change in cleavage rate (%) of porcine embryos according to the blastomere fragmentation percentage. The data are expressed as means ± SD. Different superscripts denote a significant difference (*p* < 0.05). (**D**,**E**) In porcine embryos at the cleaved stage, accumulation or intensity of adhesive junction F-actin expression (white arrows) was reduced with increasing blastomere fragment ratio. Scale bar = 50 μm. (**F**–**H**) Confirmation of changing F-actin aggregation, co-location of F-actin, and adducin fluorescence expression in Groups A, B, and C embryos. Fluorescence expressions were presented with F-actin (green), adducin (red), and nuclei (blue) using immunofluorescence and 4′,6-diamidino-2-phenylindole (DAPI) staining. Scale bar = 50 μm. Data in the bar graph are presented as the mean ± SD of three independent experiments. Differences were considered significant at * *p* < 0.05 and *** *p* < 0.001 as compared to the group A.

**Figure 2 biomedicines-10-01867-f002:**
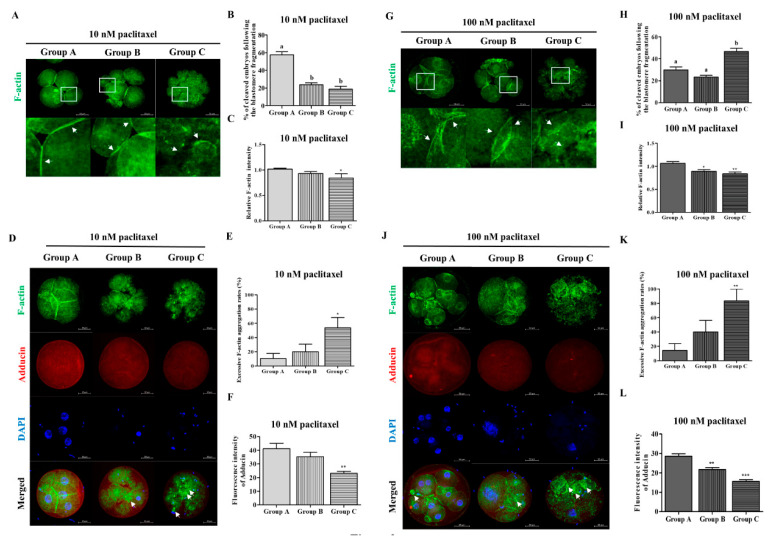
Accumulation of adhesive junction F-actin, aggregation F-actin and adducin expression by paclitaxel exposure in porcine embryos from Groups A, B, and C. (**B**,**H**) Comparison of cleaved embryos ratio from Groups A, B, and C following blastomere fragmentation according to 10 nM (**B**) and 100 nM (**H**) paclitaxel exposure in pigs. (**A**,**G**) Accumulation of adhesive junction F-actin was observed in the peri-cleavage regions (white arrows) in blastomere of cleaved embryos from Groups A, B, and C. (**C**,**I**) Fluorescent expression intensity of F-actin in 10 nM and 100 nM paclitaxel exposed embryos. Data are expressed as means ± SD. Different superscripts denote a significant difference (*p* < 0.05). (**D**–**F**,**J**–**L**) Fluorescence expressions were presented with F-actin (green), adducin (red), and nuclei (blue) using IF and DAPI staining in porcine cleaved embryos from paclitaxel treatment (10 and 100 nM). Scale bar = 50 μm. Data in the bar graph are presented as the mean ± SD of three independent experiments. Differences were considered significant at * *p* < 0.05, ** *p* < 0.01, and *** *p* < 0.001 as compared to the Group A.

**Figure 3 biomedicines-10-01867-f003:**
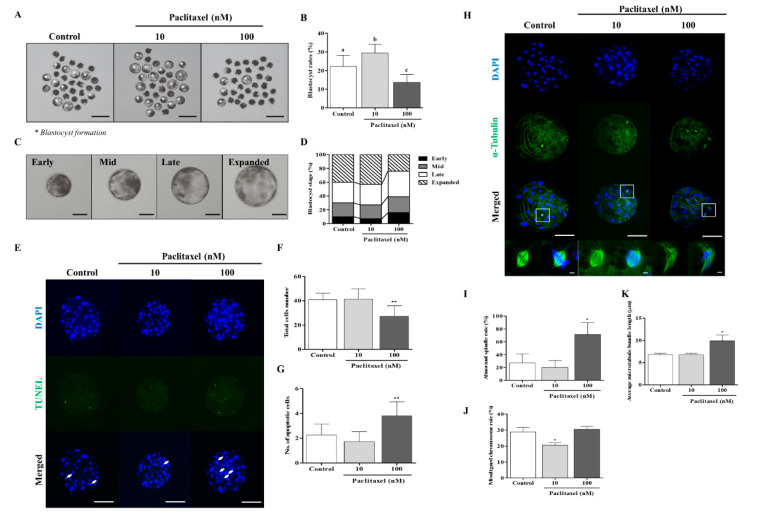
Effects of paclitaxel exposure on blastocyst developmental competence and quality in porcine embryos. (**A**,**B**) Representative photographs and total blastocyst formation number (white asterisks) by 10 and 100 nM paclitaxel exposure on porcine embryo development. Scale bar = 150 μm. (**C**,**D**) Percentage of blastocyst developmental rate as Early, Mid, Late, and Expanded stage after 10 and 100 nM paclitaxel treatment in porcine embryos. Data represent at least three independent experiments and are shown as means ± SD. Different superscript letters denote a significant difference (*p* < 0.05). (**E**–**G**) Detection of apoptotic cells and total nuclei in 10 and 100 nM paclitaxel exposed blastocysts using TUNEL (green, white arrow) and DAPI (blue) staining. The graphs show the number of total nuclei and apoptotic embryo rates in blastocysts treated with 10 and 100 nM paclitaxel. Scale bar = 50 μm. (**H**–**K**) Fluorescence expression of *α*-tubulin formation related to spindle assembly and chromosome alignment in developed porcine blastocyst after 10 and 100 nM paclitaxel treatment (*α*-tubulin: green and DAPI: blue). Data are expressed as mean ± SD and were analyzed using one-way ANOVA followed by Tukey’s multiple comparison test. Differences were considered significant at ** p* < 0.05 and *** p* < 0.01.

**Figure 4 biomedicines-10-01867-f004:**
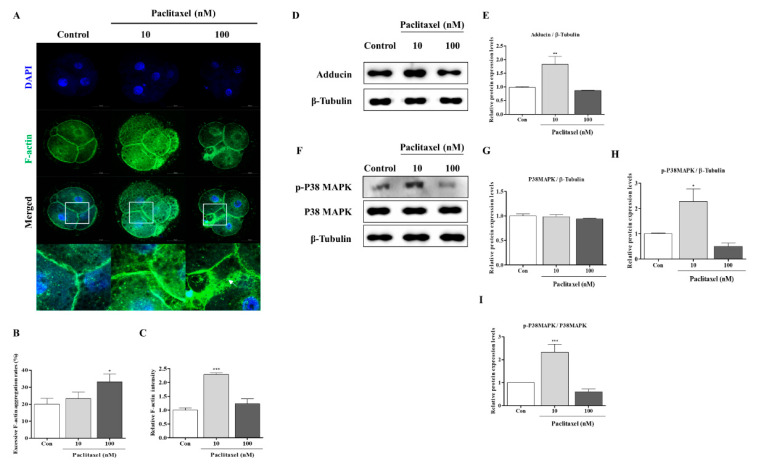
Changes of adhesive junction F-actin, P38 MAPK, and adducin protein levels in paclitaxel exposed porcine embryos at the cleavage stage. (**A**–**C**) Confirmation of adhesive junction F-actin expression by diverse paclitaxel exposure (10 and 100 nM) during the cleaved embryo stage in pigs. Fluorescence expression of F-actin (green) and DAPI (blue) in cleaved embryos treated with paclitaxel after IVC. Scale bar = 50 μm. (**D**,**E**) The protein level of adducin as an actin-binding protein in 10 and 100 nM paclitaxel exposed embryos. (**F**–**I**) Western blotting results of the P38 MAPK signal (p-P38 and P38) proteins in 10 and 100 nM paclitaxel treated embryos. Relative folds of protein levels were obtained by normalizing the signals to *β*-tubulin. Blots were probed with phosphorylation-specific antibodies for P38 MAPK and total P38 MAPK. Densitometric analysis was performed by normalizing phosphorylated P38 MAPK to total levels of P38 MAPK. Histograms represent the values of densitometry analysis obtained using ImageJ software. Data in the bar graph are presented as the means ± SD of three independent experiments (per 70 cleaved embryos). Differences were considered significant at * *p* < 0.05, ** *p* < 0.01, *** *p* < 0.001 compared to the control group.

**Figure 5 biomedicines-10-01867-f005:**
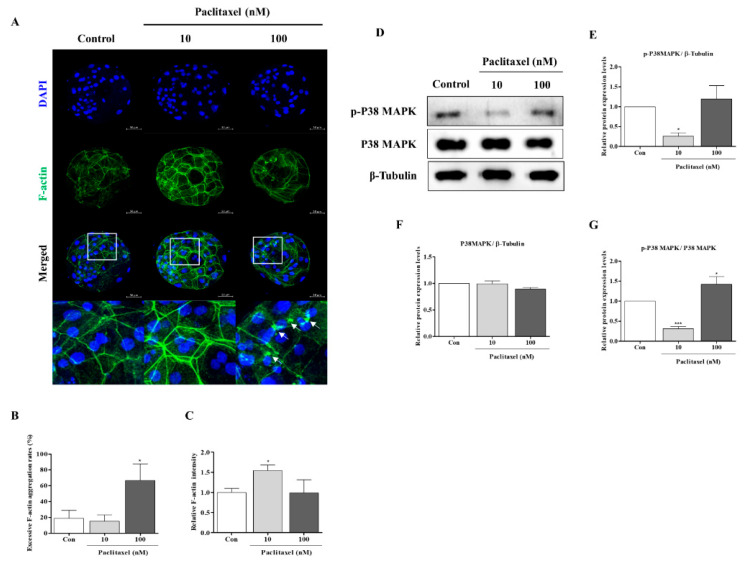
Confirmation of relationship with adhesive junction F-actin enrichment and P38 MAPK activation in developed porcine blastocysts after 10 and 100 nM paclitaxel exposure. (**A**–**C**) Changing adhesive junction F-actin expression (white arrows) by paclitaxel exposure (10 and 100 nM) at the blastocyst stage in pigs. Fluorescence expression of F-actin (green) and DAPI (blue) in the blastocyst stage treated with paclitaxel after IVC. Scale bar = 50 μm. Data on the bar graph are presented as the mean ± SD of three independent experiments. Differences were considered significant at * *p* < 0.05 compared to the control group. (**D**–**G**) Protein levels of P38 MAPK (p-P38 and P38) using Western blotting in developed porcine blastocyst from 10 and 100 nM paclitaxel treated embryos. Relative folds of protein levels were obtained by normalizing the signals to *β*-tubulin. Blots were probed with phosphorylation-specific antibodies for P38 MAPK and total P38 MAPK. Densitometric analysis was done by normalizing phosphorylated P38 MAPK to total levels of P38 MAPK. Histograms represent the values of densitometry analysis obtained using ImageJ software. Data in the bar graph are presented as the means ± SD of three independent experiments (per 50 blastocysts). Differences were considered significant at * *p* < 0.05, *** *p* < 0.001 compared to the control group.

**Figure 6 biomedicines-10-01867-f006:**
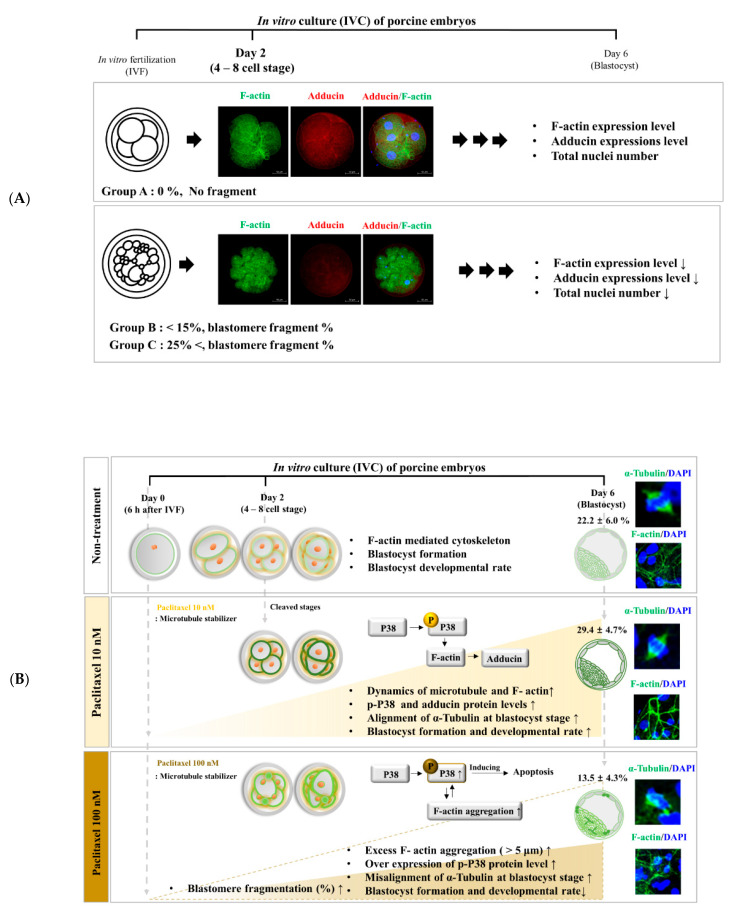
Graphical summary. Schematic diagram illustrating the correlation between F-actin/adducin cytoskeleton morphology and blastomere fragment rate related to porcine embryo quality during early embryonic development stages. ((**A**): Top panel); Embryos with blastomere fragment significantly decreased the developmental competence until blastocyst. In addition, Group A embryos showed the improving blastocyst developmental competence by F-actin and adducin related microtubule distribution and dynamics compared with Groups B and C. Simultaneously, low fluorescent expression of F-actin and adducin were observed in embryos from Groups B and C. Therefore, F-actin aggregation and adhesive junction F-actin enrichment can be suggested as a standard for evaluating the quality of embryos due to blastomere division in porcine embryos from IVC. ((**B**): Bottom panel) Abnormal cleaved blastomeres or cytokinesis defects of embryos are connected to changing F-actin distribution and enrichment at the adhesive junction site in pigs (non-treatment). Interestingly, the 10 nM paclitaxel exposed porcine embryos improve blastocyst developmental capacity through reinforced F-actin filament, adhesive junction F-actin enrichment, and increased adducin protein level in porcine embryos at the cleavage stage. Additionally, 10 nM paclitaxel also induced the p-P38 MAPK activation, improving the firmness of adhesive junction F-actin and microtubule alignment of spindle assembly at the blastocyst stage. However, 100 nM paclitaxel exposed embryos disrupted F-actin distribution, accumulated adhesive junction F-actin, and reduced adducin protein levels in porcine embryos during cleaved stages. The collapse of F-actin enrichments in embryos by 100 nM paclitaxel exposure led to the blastocyst stage with microtubule misalignment, induction of F-actin aggregation, and p-P38 MAPK over-expression. Based on these findings, we propose a positive effect of 10 nM paclitaxel for improving embryo development rate, blastocyst formation and quality via F-actin morphology, aggregation, and enrichment at the adhesive junction site of cleaving embryos in pigs.

**Table 1 biomedicines-10-01867-t001:** Cleaved embryo ratios of Groups A, B, and C according to blastomere fragmentation.

No. of Embryos Cultured	Separation Group	Fragment (%)	% of Embryos Cleaved
270	Group A	0%	58.9 ± 5.0 (159) ^a^
	Group B	<15%	23.7 ± 7.9 (64) ^b^
	Group C	>25%	17.4 ± 7.4 (47) ^b^

Data are expressed as the mean ± SD of three independent experiments. Different superscript letters indicate significant differences (*p* < 0.05).

**Table 2 biomedicines-10-01867-t002:** Percentage of Groups A, B, and C in cleaved embryos following blastomere fragmentation by paclitaxel 10 and 100 nM treatment.

Paclitaxel (nM)	No. of Embryos Cultured	Separation Group	Fragment (%)	% of Embryos Cleaved
10	315	Group A	0%	57.6 ± 9.9 ^a^
		Group B	<15%	23.6 ± 6.4 ^b^
		Group C	>25%	18.8 ± 8.8 ^c^
100	326	Group A	0%	29.8 ± 8.3 ^a^
		Group B	<15%	23.4 ± 4.6 ^a^
		Group C	>25%	46.7 ± 8.5 ^b^

Data are expressed as the mean ± SD of three independent experiments. Different superscript letters indicate significant differences (*p* < 0.05).

**Table 3 biomedicines-10-01867-t003:** Percentage of fertilization and blastocyst formation after paclitaxel treatment.

Paclitaxel (nM)	No. of Oocytes Examined	No. of Cleaved Embryos (%)	No. of Blastocysts (%)
Control	382	273 (72.0 ± 8.0)	84 (22.2 ± 6.0) ^a^
10	386	306 (78.9 ± 8.2)	111 (29.4 ± 4.7) ^b^
100	386	278 (72.8 ± 9.6)	50 (13.5 ± 4.3) ^c^

Data are expressed as the mean ± SD of three independent experiments. Different superscript letters indicate significant differences (*p* < 0.05).

**Table 4 biomedicines-10-01867-t004:** Terminal deoxynucleotidyl transferase-mediated dUTP nick-end labeling (TUNEL)-positive cell rate in blastocysts after paclitaxel treatment.

Paclitaxel (nM)	No. of Blastocysts Examined	No. of Total Cells	No. of Apoptotic Cells	Apoptosis Rate (%)
0	23	40.8 ± 5.5	2.3 ± 0.9 ^a^	5.7 ± 2.6 ^a^
10	24	41.5 ± 8.2	1.7 ± 0.8 ^a^	4.1 ± 1.5 ^a^
100	23	27.1 ± 8.8	3.8 ± 1.1 ^b^	15.4 ± 6.5 ^b^

Data are expressed as the mean ± SD of three independent experiments. Different superscript letters indicate significant differences (*p* < 0.05).

## Data Availability

Data are contained within the article.

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
