# Peer review of "Stabilization of F-Actin Cytoskeleton by Paclitaxel Improves the Blastocyst Developmental Competence through P38 MAPK Activity in Porcine Embryos"

_biomedicines, 2022, doi:10.3390/biomedicines10081867_

Round 1

Reviewer 1 Report

The study aims to test the effect of paclitaxel on the viability and stability of pig embryos.

The study is interesting and novel and it shows promising results in the elucidation of microtubule assembly mechanisms.

Several issues should be corrected, from my point of view:

L181: why are the authors clustered in three using those percentages? Based on previous work?. Otherwise please explain why you selected 15 and 25% and margins.

L187. In figures 1D and E, I cant see a dramatic reduction in F-actin intensity

L195 The authors should explain (or discuss) why the lack of differences in adducing between groups B and C, was not observed in the rest of the experiment.

L219. In experiment 3.2 a negative control 0 nM of paclitaxel is missing and it should be present. It is mentioned but I Cant find it. Please clarify.

L219 In general, I believe that the analysis is incorrect in experiments 3.2 and 3.3. From my point of view, the analysis within groups is valid, but it should be also performed between groups within doses. The authors should determine in the same group and analysis the differential effect of 0, 10, or 100 nm of PACLI, to determine its effect. Otherwise, the authors are analyzing the effect of PACLI to modify the percentage of the embryo with a given quality within groups, being the differences in the intensities produced not by the treatment, but by the embryo quality, as was demonstrated in the first experiment. For example, the analysis made in 3.4 is correct

L392 I don’t see how the results support this paragraph since no analysis of blasto rates was made on the embryos clustered by blastomeric fragmentation. Cleaved embryos are a very weak descriptor of embryo developmental competence.

L408. Similar to the previous paragraph. In addition, it seems to be that PACLI delays blasto automation FIG 3D. Did the authors examine the number of morulas in the experiment, which can develop into blastos one day later??.

L418. I think is the egg and chicken tale. F actin morphology is produced by abnormal embryo morphologies or it is the contrary. Or it is the same?. For me, actin aggregation is just an indirect m, measurement of blasto fragmentation rate.

L466 Intensity, aggregation, or both?.

L472 “control of F-actin mediated cytoskeleton and microtubule stabilization by 10 473 nM paclitaxel exposure might be useful for an evaluation basis of embryo quality”. How is that? Please clarify or rephrase.

Reviewer 2 Report

The manuscript entitled "Stabilization of F-actin cytoskeleton by paclitaxel improves the 2 blastocyst developmental competence through P38 MAPK activity in porcine embryos" investigates the effects of placlitaxel in improving the culture of porcine embryos produced by in vitro fertilization. In general terms the overall objective of the paper is somewhat confusing. Several experiments are carried out throughout the work, and the connection between them is not totally clear. The originality of the work can be qualified as average, since this product has been previously evaluated in embryos of other species as well as in porcine oocytes. In addition, the range of concentrations used is really low, and not many conclusions can be drawn. The impact of the work is also questionable. The authors found significant differences in blastocyst development rates after supplementation with 10µM, however the % of the control is around 20% when in many laboratories this same rate is usually between 30 and 40%, so the improvement proposed in the work does not represent a significant advance in the production of embryos in this species. It should also be noted that the issues raised in the manuscript hardly fit within the scope of the journal.

There are also certain details in the methodology described that need to be reviewed in depth, which I will detail below:

L-129: Define your control conditions.

L-129: According to the results described later, at this point you defined the three experimental groups in which is based part of the experiment. You should detail properly the criteria followed to establish this classification, together with the criteria used to determine the cleaved status of the zygotes. Furthermore, it is apparent from the text that the method used for classification based on the degree of fragmentation was based solely on a subjective visual observation of each presumed embryo. Despite this fact the 3 groups are established according to specific fragmentation values (<15%; >25%). The article cited for this type of classification [24] refers to the human species, giving a series of scores to the embryo based on its morphological appearance. Personally, I find this evaluation difficult to extrapolate to the porcine species, due to the particular characteristics of the porcine embryos. Can the authors detail carefully how did they stablished this protocol?

L-131: Here you state that the evaluation of the blastocysts stage was performed at day 6, however it is not explained how it was made and also the different developmental stages determined.

L-135: Describe your blocking protocol for the embryos.

L-136-137: Include the commercial references of the antibodies used. Was the cross-reactivity in the pig species evaluated?

L-144: Only the software used for the measurement is indicated here, but not the method used. In addition, three different values for expressing these fluorescence values are given in the results: Relative F-actin intensity (normalizing the group A to 1 and expressing the rest of the groups relative to it), excessive F-actin aggregation (a concept which is not fully defined at all in the whole manuscript) and fluorescence intensity of Adducin (which is expressed as an absolute value but not a relative as for the F-actin). At this point the criteria should be unified and adequately explained in the corresponding section.

L-154: Explain the ratios and percentages obtained with the TUNEL technique at this point.

L-157: More details should be provided about the Western Blot protocol. Detail the number of cleaved embryos/blastocysts used per sample. Did you check the amount of total protein isolated from each sample? Why was β-tubulin chosen as a reference protein? According to the experimental design a protein involved with cytoskeletal function would not be the best option to run the WB. Did you test the specificity of the antibodies used using blocking peptides? In the WB pictures it can not be seen the ladder to check the molecular weight, you should include this at the images.

L-171: The manuscript says that both, SD and SEM were used to express the deviation of the different data. You should only use one, preferably the SD.

L-172: All the statistical tests performed are specific for normal or parametric distributions of the samples. Did you check the normality of your different parameters analyzed?

L-177: The different sections of the results are preceded by some kind of justification for the test. I think this should be presented either in the discussion, or reorganized in some other way, since this section should present only the results.

For these various reasons I consider that the manuscript is not suitable for publication in its present state, so my recommendation is to reject it.

Reviewer 3 Report

In this manuscript, the authors use porcine embryos to evaluate the effect of two concentrations on paclitaxel on cytoskeleton morphology. The authors conducted several experiments including investigating the relationship between adhesive junction F-actin enrichment and P38 MAPK activation. Overall, the data is well presented, and the conclusions are supported by the results. The introduction provides sufficient background, and the materials generally provides the necessary details for the experiments.

My primary suggestion is to potentially make a few parts of the manuscript more concise to improve the presentation. Tables 2 and 3 are largely redundant and could be combined into a single table without loss of information. The discussion, while well written, repeats much of the results section. It was a bit difficult to discern what was new in the discussion and what repeated the observations and points from the results. The discussion could be made more impactful if focused on extensions of the results section.

Minor points:

Line 115 – if the pH of the Tris-buffered solution is known, it can be stated (and the same for all buffers)

Line 282 – states “To prove…”, the experiment does not “prove” but indicates or suggests

Line 467 – the word “the” can be omitted

Lines 468-470 – the sentence is not clear

Reviewer 4 Report

The manuscript Joe and coworkers entitled: Stabilization of F-actin cytoskeleton by paclitaxel improves the blastocyst developmental competence through P38 MAPK ac- 3 tivity in porcine embryos describe that paclitaxel (Taxol) used well known for several  anti cancer therapies is able to stabilize F-actin through p38 MAPK activation. For this work was performed with blastocysts from pig embryos. The work gave some interesting new results in the actin research and due to the nice pictures except some required minor revision is the work well done. The influence of paclitaxel is visible. Pigs are often used for blood pressure research, whereas adducin is known bind to ends of actin filaments, since actin and spectrin,  can form a two-dimensional network of pentagonal and hexagonal structures below the cell membrane. It is unclear which adducin was shown and also which F-actin is the observed form. Also is known that adducin is involved in the binding of the α-subunit of Na+/K+-ATPase with the spectrin-actin membrane cytoskeleton. The activity of adducin depends on its degree of phosphorylation (by protein kinase C and A) and the intracellular Ca+ concentration. This should be excluded as a control experiment, that we do not see this pathway. 

In the discussion a citation must be included Iwamoto, T. & Kita, S. (2006): Hypertension, Na+/Ca2+ exchanger, and Na+/K+ ATPase. In: Kidney Int. Bd. 69, S. 2148–2154 and  Manunta, P. et al. (2006): A new antihypertensive agent that antagonizes the prohypertensive effect of endogenous ouabain and adducin. In: Cardiovasc. Hematol. Agents. Med. Chem. Bd. 4, S. 61–66. PMID 16529550

Which adducin form is seen and which actin is responsible for the network?

A commercial actin in vitro polymerization assay can help to support the paclitaxel effect on actin.  

In the experimental section complete details to the materials must added, e.g., (P5282, company city country) through the section is to improve.

A scheme how the data are generated could be helpful

Round 2

Reviewer 1 Report

.

Reviewer 4 Report

After this intense revision the manuscript can be accepted